# A Study on the Factors Influencing Smoking in Multicultural Youths in Korea

**DOI:** 10.3390/healthcare11101437

**Published:** 2023-05-15

**Authors:** Jin-Hee Park, Mi-Jin Kim, Hee-Joo Lee

**Affiliations:** 1Department of Nursing, Changshin University, 262 Palyongro, MasanHoewon-gu, Changwon 51352, Gyeongsangnam-do, Republic of Korea; parkjh124@cs.ac.kr; 2Department of Nursing, Daegu Haany University, Hanuidae-ro, Gyeongsan-si 38610, Gyeongsangbuk-do, Republic of Korea; 3Department of Nursing, Sangmyung University, 31 Sangmyungdae-gil, Dongnam-gu, Cheonan 31066, Chungnam, Republic of Korea; foremost@smu.ac.kr

**Keywords:** adolescent, smoking, cultural diversity, community, ecological

## Abstract

Based on the ecological integration model, this study examined the factors affecting smoking in adolescents from multicultural families by dividing them into two levels: microsystem and social network factors. The data were from the Korea Youth Risk Behavior Web-Based Survey (KYRBS) from 2016 to 2020. It included 4577 respondents whose fathers, mothers, or both, were not born in Korea. The factors affecting smoking among multicultural teenagers were determined by a composite-sample multiple logistic regression analysis. Male smoking rates among multicultural adolescents were 2.49 times higher than female rates in the microsystem. When the father was “Korean” rather than a “Foreigner”, smoking was 0.55 times lower in family factors in terms of social network. In social factors of social networks, multicultural adolescents’ smoking was 12.02 times greater when they were drinking than when they were not, and 3.62 times higher when the answer to the question of whether they had experienced violence was “yes” than “no.” Based on the ecological model in this study, social factors such as drinking, and violence were highly related to smoking. Since multicultural adolescents were closely influenced by the surrounding environment, such as family, school, and social relationship, it was necessary to let parents and schoolteachers be involved in the intervention of smoking of multicultural adolescents so that they can help multicultural adolescents adjust better to school and perform better academically while decreasing risky behaviors for their health, such as drinking and, ultimately, smoking.

## 1. Introduction

### 1.1. Background of the Study

As the influx of people from multicultural backgrounds into Korean society increases, the number of children from multicultural families will increase further in the future, considering the recent trend of middle-entry immigrant youth and foreign workers’ children [1]. The highest percentage of youth from multicultural families are children of families of international marriages that are born and raised in Korea, and the characteristics of children vary widely depending on the background of the multicultural families [1].

In a study that examined the health risk behaviors of adolescents in multicultural families, families with Korean parents, and families with North Korean parents, adolescents from multicultural families showed higher health risk behaviors, such as smoking, drinking, drug use, and sex, than adolescents from families with Korean parents [2]. In addition, the health risk behavior of teenagers from families with North Korean parents was reported higher than those of other multicultural families, and teenagers from multicultural families had more mental problems, such as depression, stress, and suicide attempts, than those from families with Korean parents [2]. According to a study on the health behavior of teenagers from North Korean families, multicultural families, and Korean families, smoking and drinking rates of teenagers from North Korean families are more than twice as high as those from South Korean families [3], indicating that health risk behavior differs depending on the type of multicultural families.

In a study on the effect of smoking on adolescents’ health, it was reported that the quality of sleep was very low, and the group of adolescents that used both general and e-cigarettes did not get enough sleep [4]. In addition, it was noted that the experience of depression, stress perception, and subjective sleep dissatisfaction, which are characteristics of mental health, affect smoking in adolescents [5]. Moreover, the mental health problems of adolescents’ depression, stress, suicidal thoughts, and lack of sleep often lead to health risk behaviors such as drinking and smoking [6]. Two out of ten teenagers from multicultural families in Korea experienced smoking and suicidal thoughts, and about one in three experienced depression, and the experience of smoking of teenagers from multicultural families was reported to have a significant positive effect on suicidal thoughts [7]. It was also reported that smoking, drinking, and drug use were high among immigrants with depression and anxiety, including intermediate-entry immigrants [8].

Nicotine, the main ingredient of cigarettes, is known to be highly addictive, and if people start smoking in adolescence, they often continue to smoke until they become old. In fact, this addiction is known to be very difficult to quit when people start in younger age. The younger one starts smoking, the harder it is to quit [9]. Accordingly, there is a need for an active prevention policy to inform the public of the risk of smoking in adolescence in advance and to prevent teenagers from accessing nicotine in cigarettes [9]. In addition, smoking in adolescence often acts as a so-called “gateway drug” that leads to drug abuse, and smoking adolescents are significantly more likely to begin drinking and abusing drugs than non-smoking adolescents. In addition, smoking in adolescence is closely related to causing mental problems, such as depression, stress, and anxiety, and exposing them to situations such as an unsafe sex life, becoming a runaway, and school violence [10].

From the studies of above, smoking among adolescents from multicultural families is highly related with their mental and physical risks, such as depression, stress, sleep deprivation, drinking and even using illicit drugs. Therefore, research on factors that influence behavior by identifying objective characteristics of youth from multicultural families will be used as important data to establish policies for youth from multicultural families and open them to a safer future society.

### 1.2. Conceptual Framework

Among theories related to human behavior, the ecological system theory is a theory that considers the internal and external aspects of an individual and focuses on the interaction between an individual and the various environmental systems surrounding it [11]. The ecological integration model explains that the interaction between systems affects individual behavior by dividing the environment that affects individual behavior into microsystems, social networks, institutional system, and macrosystem. In theory, microsystem refers to the role of an individual in an environment that directly surrounds the individual as a personal characteristic variable, and social network refers to interpersonal relationships between friends, family, and neighbors. Institutional system is a variable of organizational characteristics, including microsystem, social network, and institutions to which individuals belong, and macrosystem is a socio-cultural variable, which refers to the policies, laws, etc. of the society or institutions to which individuals belong. Smoking in adolescents from multicultural families depends on the characteristics of individuals born in multicultural families.

It can be inferred that it may be influenced by the family and social environment surrounding the individual. Therefore, applying the ecological integration model to explain the factors of smoking in youth from multicultural families, personal factors, social network factors, and the interactions between them can be used to explain the effects of smoking in youth from multicultural families.

Based on the ecological integration model and previous studies, this study examined the factors influencing smoking in adolescents from multicultural families by dividing them into two levels: micro and social network factors. This study arranged a microsystem of factors influencing smoking in multicultural adolescents, comprising gender, grade, school achievement, sleep, stress, depression and suicidal thought as personal level variables; nationality of parents, residence type and economic status of multicultural adolescents as family factors of a social network; and drinking, drug use and violence as social factors of a social network. Based on the above, the conceptual framework of this study is presented as shown in Figure 1.

### 1.3. Purpose of the Study

Based on the ecological integration model, this study examined the factors affecting smoking in adolescents from multicultural families by dividing them into two levels: microsystem and social network factors. It is believed that the interaction between these levels can explain the factors that affect smoking in adolescents from multicultural families. In addition, grasping the objective characteristics of youth from multicultural families and understanding the factors affecting behavior are believed to be useful as basic data to open policy establishment for youth from multicultural families and integration into a safer future society.

The study’s specific purposes were as follows: (1) to identify the microsystem, social network, and the number of smoking adolescents from multicultural families; (2) to explore the difference in smoking among adolescents from multicultural families according to the microsystem and social network; and (3) to identify the factors influencing smoking among adolescents from multicultural families.

## 2. Materials and Methods

### 2.1. Study Design

This study uses a cross-sectional research design and an ecological system theory approach to identify the factors influencing smoking among multicultural adolescents. It uses raw data from the Korea Youth Risk Behavior Web-based Survey (KYRBS) from 2016 to 2020 [12].

### 2.2. Data Collection

The KYRBS has been conducted annually since 2005 with the aim of understanding the current status and trends of health behaviors of teenagers in Korea. An online survey was administered to anonymous volunteers across the country who were adolescents from the first year of middle school to the last year of high school. The sampling process was divided into stages for population stratification, sample allocation, and sample extraction. For sampling, we used a stratified collection extraction method, in which the first extraction unit was the school’s permanent random number and the second extraction unit was randomly selected as a class. In the KYRBS, the raw data included logical errors and outliers, and weights were calculated by the Korea Disease Control and Prevention Agency and provided as raw data weighting variables. The weights were multiplied by the weights’ posterior correction rate multiplied by the extraction rate reciprocal and response rate reciprocal, and the weights by region, gender, school type (middle, general, specialized school) and grade were calculated to be equal to the number of middle and high school students nationwide by year. Responses to the online survey about youth health behavior were organized using unique numbers that could not be used to determine the respondents’ identities; thus, no personal information was disclosed, and anonymity and confidentiality were guaranteed.

### 2.3. Participants

Raw data from the Korea Youth Risk Behavior Web-based Survey (KYRBS) from 2016 to 2020 [12], conducted annually by the Korea Centers for Disease Control and Prevention, were used. Of the 300,096 participants in the KYRBS from 2016 to 2020, 4577 of them answered “no” to the question “Was your father born in Korea?”, or “Was your mother born in Korea?” or “Were your both mother and father born in Korea?” Therefore, these 4577 respondents were our study participants.

The researchers conducted the study after receiving a review exemption from Daegu Haany University with which the corresponding author was affiliated (IRB No: 2023-1-03).

### 2.4. Tools

The study’s dependent variable was multicultural smoking adolescents, whereas the independent variables were the microsystem (personal factors) and social network (family factors, social factors).

### 2.5. Microsystem

The microsystem refers to the personal factors of multicultural adolescents and comprises gender, grade, school achievement, sleep, stress, depression, and suicidal thought. Gender could be answered with either “male” or “female”; grade with “middle school” or “high school”; and school achievement with “high,” “middle,” or “low”. Sleep was answered with “very sufficient,” “sufficient,” “moderate,” “insufficient,” or “not sufficient at all” to the question, “In the last 7 days, do you think the amount of sleep you have slept is sufficient to recover from fatigue?” Stress was answered with “too much,” “much,” “a little,” “not much,” or “not at all” to the question, “How much stress do you usually feel?” Depression was answered with “yes” or “no” to the question, “In the last 12 months, have you felt sad or hopeless enough to stop your everyday life for 2 weeks?” Suicidal thought was answered with “yes” or “no” to the question, “In the last 12 months, have you thought about suicide?”

### 2.6. Social Network

The social network includes family factors and social factors of multicultural adolescents. Family factors comprises the father’s and mother’s nationality, residence type, and economic status. The father’s and mother’s nationality were classified as “Foreigner” or “Korean”; residence type was classified as “with family,” “boarding/self-boarding,” “with relatives,” “dormitory,” and “child-care”; and economic status was classified as “high,” “moderate,” and “low.” Social factors included experiences in drinking, drug use, and violence of multicultural adolescents. Drinking was answered with either “yes” or “no” to the question, “Have you had more than one drink so far?” Drug use was answered with either “yes” or “no” to the question, “Have you ever habitually or deliberately taken drugs or drank butane gas or bond so far?” Violence was answered with “yes” or “no” to the question, “In the past 12 months, have you been treated at a hospital because of violence (physical assault, intimidation, bullying, etc.) by a friend, senior, or adult?”

### 2.7. Smoking

Smoking was answered with either “yes” or “no” to the question which was “Have you ever taken one or two drags from a cigarette?”

### 2.8. Data Analysis

The data used in this study were evaluated using composite sample analysis that reflected strata, cluster, weight (W), and a finite population modifications coefficient. Data were analyzed using IBM SPSS Statistics 23. The following were analyzed:The frequency and percentage of the microsystem and social network, and smoking of multicultural adolescents.The differences in smoking according to the microsystem and social network among multicultural adolescents by using the Rao–Scott chi-square test.The factors affecting smoking in multicultural adolescents using multiple logistic regression analysis.

## 3. Results

### 3.1. Differences in Smoking according to Multicultural Adolescents’ Microsystem and Social Network

As shown in Table 1, it was found that 620 (13.5%) of multicultural adolescents smoked. In terms of the microsystem, smoking consisted of more males than females (404 (65.2%) vs. 216 (34.8%)), more high school adolescents smoked than middle school adolescents (358 (57.7%) vs. 262 (42.3%)), and the smoking rate was highest when school achievement was low (297 (47.9%)). The adolescents found that they smoked more when they had moderate sleep satisfaction (187 (30.2%)) and were a little stressed (219 (35.3%)). They found themselves smoking more when they were not depressed (367 (59.2%)) and had no suicidal thoughts (457 (73.7%)).

Among the family factors in the social network, the father’s nationality, when the father was Korean there smoking was 349 (56.3%) vs. when their father was foreigner smoking was 225 (36.3%). The smoking rate was highest were residence type was with family (498 (80.3%)) and when economic status was moderate (227 (36.6%)). Among the social factors in social networks, smoking was higher when they were drinking (518 (83.5%)), not using drugs (617 (99.5%)), and having no experience of violence (491 (79.2%)). Table 1 presents the results.

### 3.2. Factors Influencing Smoking among Multicultural Adolescents

Table 2 shows the results obtained from the multiple logistic regression analysis to identify the factors influencing smoking among multicultural adolescents. We considered the statistically significant variables of the microsystem, family factors, and social factors of the social network. We analyzed these factors using multiple logistics regression and found that the resulting model was significant (Wald F = 23.055, *p* < 0.001). The explanatory power was 41.5%. The factors influencing smoking among multicultural adolescents were microsystem (gender), and family factors of social network (father’s nationality), social factors of social network (drinking and violence).

Among the microsystem, multicultural adolescents’ smoking was 2.49 times higher in males than in the female. In family factors in terms of the social network, smoking was 0.55 times lower when the father’s nationality was “Korean” compared to “foreigner”. Among the social factors in terms of the social network, multicultural adolescents’ smoking was 12.02 times higher when they were drinking than when they were not drinking, and 3.62 times higher when the answer to violence was “yes” compared “no”. These results are shown in Table 2.

## 4. Discussion

Based on the ecological integration model, this study examined the factors affecting smoking among multicultural adolescents from the subjects of the Korea Youth Risk Behaviour Web-Based Survey (KYRBS) for five years from 2016 to 2020. The ecological model states that the environment and relationships of various dimensions surrounding an individual affect each other, which affects the development of individual behavior [13]. In this study, among multicultural adolescents from 2016 to 2020, in terms of microsystem, male students were 2.49 times more likely to smoke than female students, and in the family factors of the social network, if their fathers were born in Korea, they were 0.55 times less likely to smoke than foreign fathers. In addition, while they were drinking, they were 12.02 times more likely to smoke, and if they experienced violence, they were 3.62 times more likely to smoke compared to no violence. In fact, the social factors in the social network, such as drinking, and violence significantly affected smoking in multicultural adolescents. In addition, in this study, it was found that drinking was the highest related factor affecting smoking among multicultural adolescents.

Of the 4577 respondents, 620 (13.5%) said they smoked, 1496 (32.7%) said they drank, and 518 (83.5%) said they smoked and drank. According to a report that studied the health behavior of 759 multicultural adolescents in 2016, 123 (19.3%) said they smoked, and 247 (48.2%) said they drank [14]. In 2021, 5 out of 100 middle and high school students in Korea (4.5%) reported smoking in the last 30 days, and 11 (10.7%) reported drinking in the last 30 days [15], so the smoking and drinking rate of multicultural adolescents is higher than that of middle and high school students in Korea.

In Korea, even though the sale of cigarettes and alcohol to adolescents under the age of 19 is prohibited, many adolescents smoke and drink, and multicultural adolescents are more likely to engage in health-risk activities because their social environment is more vulnerable. It was reported that adolescents from multicultural families are more likely to be exposed to the risk of smoking than drinking [16].

It was reported that the smoking experience of multicultural adolescents was significantly higher in male students than in female students. Smoking among multicultural adolescents is often accompanied by health risk behaviors, such as drinking, which had a significant correlation between their first time smoking and drinking [14]. It is believed that smoking prevention education is necessary from a young age.

In this study, social factor that affected the smoking of multicultural adolescents was violent experiences, and 233 (5.1%) of the subjects said they had experienced violence. In a report that studied the experiences of violence among multicultural adolescents, adolescents from North Korean parents, and adolescents from Korean parents, adolescents from North Korean families were 11 times more exposed to violence, and multicultural adolescents were 5 times more exposed than adolescents from Korean parents [17].

It has been reported that experiences of violence damage among multicultural adolescents has a statically significant effect on current smoking behavior, and the higher the frequency of violence damage, the higher the likelihood of smoking behavior [18]. As a result of this study, the experience of violence among multicultural adolescents is related to their smoking, and health risk behaviors such as smoking, and drinking are accompanied.

The frequency of smoking and drinking among adolescents in Korea is increasing, and smoking is often accompanied by drinking; smoking and drinking in adolescence are major social problem in Korea because it affects them even in adulthood [19]. According to a report that studied smoking by adolescents with an ecological model, individual perceptions, attitudes, relationships with friends, and parents’ attitudes toward smoking influences smoking rates of adolescents. Parents’ smoking has a great influence on teenagers living together in the family, and it is said that smoking is less common in adolescents who do not smoke and live with their parents [20]. According to a report that studied the smoking and drinking behavior of Korean adolescents using another ecological model, male adolescents were more likely to smoke and drink than female adolescents due to personal factors. Since the family relationship factor is a basic unit for adolescents to learn social studies, it is said that the teaching of parents, who are role models of adolescents, has a very significant influence on youth smoking and drinking [19].

Based on the ecological integration model, it is reported that gender in the microsystem, the father’s nationality; in the family factors of the social network, drinking and violence; and the social factors of the social network affect multicultural adolescents smoking, and they are indicating that the social environment surrounding multicultural adolescents is highly related to smoking. It was found that multicultural adolescents were closely influenced by their surrounding environment, such as family, school, and social relationships, and parents had a strong influence on their adolescent’s health behavior, so it was necessary to explore parents’ health-related characteristics [21].

Health experts, including health teachers, suggested that smoking, which forms the identity of the youth’s peer group, should be resolved in a constructive direction, similar to music and club activities, specifically through group smoking cessation counseling. In addition, it was said that educational intervention on socio-technical factors, such as communication skills or self-assertion training, can be properly rejected in situations where smoking is encouraged or pressured [22]. Furthermore, among the smoking cessation programs for teenagers, it was reported that the smoking cessation effect was the most effective when receiving school-based programs and intensive treatment by smoking cessation education experts [23]. Therefore, to prevent smoking among multicultural adolescents, it is considered necessary to have special programs and activities belonging to the community as well as regular classes in schools.

Smoking among multicultural adolescents is a dangerous act that is accompanied by drinking, which is expected to affect them even after they become adults in the future. As a result, policies, and intervention methods to prevent smoking among multicultural adolescents are urgently needed, and an integrated community intervention program that includes not only multicultural adolescents but also parents, friends, and schoolteachers from multicultural families is needed.

Although Korean adults’ superficial understanding of multiculturalism has increased, multicultural acceptance is segmented, biased preferences and discrimination are intensifying, and the anti-multicultural consciousness of the younger generation is being strengthened. In addition, Koreans’ perceived sense of threat of foreigners is deepening, and multicultural education is insignificant. This suggests that Korea’s policy on multicultural acceptance is necessary for a society that lives together [24].

In the process of youths from multicultural families being a part of our society, the more direct experiences of biculturalism, the higher the bicultural acceptance attitude and multicultural acceptance, so it is believed that awareness improvement education that can increase multicultural understanding and acceptance attitude should be conducted [25]. It was reported that multicultural adolescents’ positive attitudes toward Korean culture and foreign culture had a positive effect on school adaptation, and it was suggested that mediation and education to form their identity were necessary to increase their adaptation to school [26]. Multicultural adolescents will be an important human resource in Korean society in the future, and their health is considered to have a great impact on Korean society, so it is necessary to accept and pay attention to friends, parents, schools, and local communities.

As was previously indicated, behaviors acquired in adolescence have a lasting impact on young people’s lives and can persist far into late adulthood. Thus, it’s crucial to develop healthy habits during adolescents. The demand for psychological and mental care is rising as quickly as the number of multicultural adolescents both at home and abroad. Supporting multicultural youth results in better school adaption because of positive peer relationships and after-school programs. Additionally, it promotes parental involvement and consideration in general.

Regarding the study’s limitations, the self-report and extensive raw data were both analyzed using weights. Multicultural adolescents who today have different cultures, nations, and levels of Korean language proficiency cannot, therefore, be generalized in this study. Future studies should therefore concentrate on variables that influence the smoking rate of multicultural teenagers considering variations in culture, nationality, and Korean language proficiency.

Regarding this study’s limitations, this study included large-scale raw data and self-reports. Hence, this study cannot be generalized because it does not consider the timing of adolescent’s smoking start, whether they have attempted to quit smoking, or their interactions with peers.

## 5. Conclusions

Based on the ecological integration model, this study examined the factors affecting smoking among multicultural adolescents from the subjects of the Korea Youth Risk Behaviour Web-Based Survey (KYRBS) for five years from 2016 to 2020. Among 4577 multicultural adolescents from 2016 to 2020, in terms of the microsystem, male students were 2.49 times more likely to smoke than female students, and in the family factors of the social network, if their fathers were born in Korea, they were 0.55 times less likely to smoke than those with foreign fathers and 12.02 times more likely to experience violence in social factors in terms of the social network. In addition, in this study, it was found that drinking was the highest related factor affecting smoking among multicultural adolescents.

Therefore, it is important to think about educational and psychological solutions, which call for a multifaceted strategy involving parents, peers, and teachers. These interventions help multicultural adolescents adapt better to school, experience less drinking, and eventually smoke less, leading to lower rates of these risky behaviors.

## Figures and Tables

**Figure 1 healthcare-11-01437-f001:**
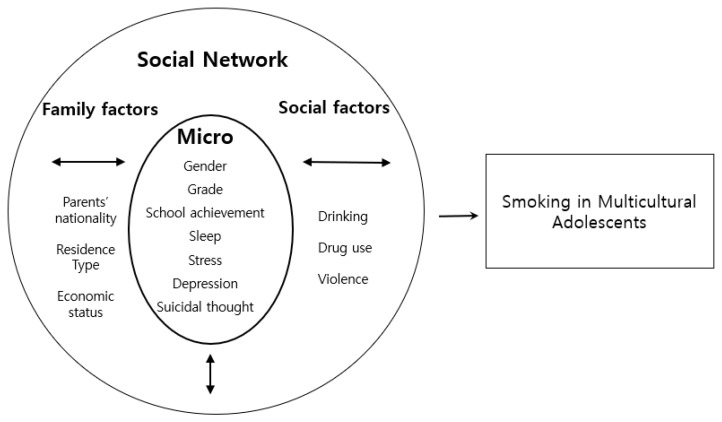
The conceptual framework of this study.

**Table 1 healthcare-11-01437-t001:** Differences in smoking according to multicultural adolescents’ general characteristics, health risk behaviors, mental health (*N* = 4577).

Characteristics	Categories	* *n* (%)	Smoking		Modified *F* ^†^	df1 (df2)	*p*
			Yes * *n* (%)	No * *n* (%)			
Smoking		4577 (100.0)	620 (13.5)	3957 (86.5)			
1. Microsystem							
Gender	Male	2191 (47.9)	404 (65.2)	1787 (45.2)	61.58	1 (1791)	<0.001
	Female	2386 (52.1)	216 (34.8)	2170 (54.8)			
Grade	Middle school	2937 (64.2)	262 (42.3)	2675 (67.6)	106.98	1 (1791)	<0.001
	High school	1640 (35.8)	358 (57.7)	1282 (32.4)			
School achievement	High	1375 (30.0)	198 (31.9)	1177 (29.7)	13.71	1.99 (3572.3)	<0.001
	Middle	1323 (28.9)	125 (20.2)	1198 (30.3)			
	Low	1879 (41.1)	297 (47.9)	1582 (40.0)			
Sleep	Very sufficient	460 (10.1)	41 (6.6)	419 (10.6)	18.91	3.96 (7100.8)	<0.001
	Sufficient	964 (21.1)	90 (14.5)	874 (22.1)			
	Moderate	1531 (33.4)	187 (30.2)	1344 (34.0)			
	Insufficient	1086 (23.7)	169 (27.3)	917 (23.2)			
	Not sufficient at all	536 (11.7)	133 (21.5)	403 (10.2)			
Stress	Too much	539 (11.8)	105 (16.9)	434 (11.0)	11.57	3.99 (7141.1)	<0.001
	Much	1191 (26.0)	145 (23.4)	1046 (26.4)			
	A little	1860 (40.6)	219 (35.3)	1641 (41.5)			
	Not much	786 (17.2)	102 (16.5)	684 (17.3)			
	Not at all	201 (4.4)	49 (7.9)	152 (3.8)			
Depression	Yes	1259 (27.5)	253 (40.8)	1006 (25.4)	49.61	1 (1791)	<0.001
	No	3318 (72.5)	367 (59.2)	2951 (74.6)			
Suicidal thought	Yes	666 (14.6)	163 (26.3)	503 (12.7)	70.41	1 (1791)	<0.001
	No	3911 (85.4)	457 (73.7)	3454 (87.3)			
2. Social network							
(1) Family factors							
Father’s Nationality ^‡^ (*n* = 4294)	Korean	3297 (72.0)	349 (56.3)	2948 (74.5)	82.31	1 (1791)	<0.001
	Foreigner	997 (21.8)	225 (36.3)	772 (19.5)			
Mother’s Nationality ^‡^ (*n* = 4480)	Foreigner	295 (6.4)	43 (6.9)	252 (6.4)	1.67	1 (1791)	0.196
	Korean	4185 (91.4)	529 (85.3)	3656 (92.4)			
Residence type	With family	4210 (92.0)	498 (80.3)	3712 (93.8)	30.28	3.97 (7117.7)	<0.001
	Boarding/ self-boarding	65 (1.4)	22 (3.5)	43 (1.1)			
	With relatives	111 (2.4)	37 (6.0)	74 (1.9)			
	Dormitory	147 (3.2)	44 (7.1)	103 (2.6)			
	Child-care	44 (1.0)	19 (3.1)	25 (0.6)			
Economic status	High	1135 (24.8)	169 (27.3)	966 (24.4)	20.10	2.00 (3576.9)	<0.001
	Moderate	2244 (49.0)	227 (36.6)	2017 (51.0)			
	Low	1198 (26.2)	224 (36.1)	974 (24.6)			
(2) Social factors							
Drinking	Yes	1496 (32.7)	518 (83.5)	978 (24.7)	593.56	1 (1791)	<0.001
	No	3081 (67.3)	102 (16.5)	2979 (75.3)			
Drug use	Yes	41 (0.9)	3 (0.5)	38 (1.0)	1.41	1 (1791)	0.235
	No	4536 (99.1)	617 (99.5)	3919 (99.0)			
Violence	Yes	233 (5.1)	129 (20.8)	104 (2.6)	298.31	1 (1791)	<0.001
	No	4344 (94.9)	491 (79.2)	3853 (97.4)			

* *n* is the unweighted sample size and percent (%) is weighted percent, which is calculated by complex sample analysis; ^†^ Modified *F*, calculated by complex sample analysis. ^‡^ Different from total number (*n* = 4577) due to missing values, which were no response to father’s and mother’s nationality.

**Table 2 healthcare-11-01437-t002:** Factors Influencing Smoking among Multicultural Adolescents (*N* = 4577).

Characteristics	Categories	*B*	OR	95% CI			*p*
1. Microsystem							
Gender (ref. Female)	Male	0.911	2.49	1.93	-	3.21	<0.001
Grade (ref. High school)	Middle school	−0.340	0.71	0.55	-	0.92	0.010
School achievement (ref. Low)	High	−0.297	0.74	0.56	-	0.98	0.036
	Medium	−0.465	0.63	0.47	-	0.85	0.002
Sleep (ref. Not sufficient at all)	Very sufficient	−0.745	0.48	0.27	-	0.82	0.008
	Sufficient	−0.445	0.64	0.42	-	0.99	0.044
	Normal	−0.521	0.59	0.41	-	0.86	0.005
	Insufficient	−0.222	0.80	0.56	-	1.14	0.216
Stress (ref. Not at all)	Too much	−0.597	0.55	0.30	-	1.02	0.059
	Much	−0.583	0.56	0.32	-	0.98	0.042
	A little	−0.662	0.52	0.30	-	0.88	0.014
	Not much	−0.228	0.80	0.45	-	1.41	0.437
Depression (ref. No)	Yes	0.128	1.14	0.84	-	1.53	0.400
Suicidal thought (ref. No)	Yes	0.542	1.72	1.21	-	2.44	0.002
2. Social network							
(1) Family factors							
Father’s nationality (ref. Foreigner)	Korean	−0.595	0.55	0.42	-	0.72	<0.001
Residence type (ref. Child−care)	With family	0.599	1.82	0.68	-	4.87	0.232
	Boarding /Self−boarding	0.806	2.24	0.73	-	6.90	0.160
	With relatives	1.188	3.28	1.00	-	10.73	0.049
	Dormitory	0.780	2.18	0.71	-	6.67	0.171
Economic status (ref. Low)	High	−0.182	0.83	0.61	-	1.15	0.261
	Moderate	−0.333	0.72	0.53	-	0.97	0.028
(2) Social factors							
Drinking (ref. No)	Yes	2.486	12.02	9.20	-	15.70	<0.001
Violence (ref. No)	Yes	1.286	3.62	2.28	-	5.75	<0.001

Nagelkerke R2 = 0.415, Cox and Snell R2 = 0.239, Wald F = 23.055, *p* < 0.001; CI, Confidence interval; OR, odds ratio; ref., reference.

## Data Availability

The data presented in this study are available on request from the corresponding author. The data are not publicly available due to privacy restrictions.

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
