# Peer review of "A Study on the Factors Influencing Smoking in Multicultural Youths in Korea"

_healthcare, 2023, doi:10.3390/healthcare11101437_

Round 1

Reviewer 1 Report

This study examines the Factors Influencing Smoking in Multicultural Youths in Korea: Using Statistics from the Korea Youth Risk Behavior Web-based Survey (KYRBS) (2016-2020). This is a good addition to the literature in the field of human behavior and health. Several issues need to be corrected or clarified to warrant its publication.

 Abstract:

The conclusion that there are factors that affect affecting adolescents from multicultural families is not postulated in the conclusion. The authors only mention that education and psychological interventions are required, but how this is related to the actual study? While there are factors that contribute to smoking behavior, what are those?

Introduction:

Smoking in adolescents from multicultural families can cause mental and physical risks, so it is important to identify the factors that influence smoking in adolescents from multicultural families. How do those factors influence tobacco smoke? There are various factors indicated by the authors, but none was correlated to smoking. What kind of cigarettes are consumed? Brands? Since this is from 2016 to 2020, does the survey consider vaping? Electronic cigarettes?

The introduction is too long and most of the information should be moved to the methods section.

There are three purposes of the study: How the microsystem was identified? How the factors that influence smoking was determined?

What are the ages of the participants? Because it is not the same compared to a teen of 16 to 14 years of age. How was corrected for missing data? The introduction indicated, "According to a study on the health behavior of teenagers from North Korean families, multicultural families, and Korean families, smoking and drinking rates…”, but Table 1 social factors doesn’t reflect this statement.

 Table 1 should be separated into their respective categories. It is hard to read and interpret.

Table 1, school achievement doesn’t correlate with Table 2 grade (ref High School), what about other grades middle school, dropping school

Any reports that indicate if they (teens) attempt to quit?

Other factors like income, housing, unemployed (parents), nicotine exposure during childhood, parental and peers’ example of smoking, nicotine dependence, awareness of smoking,

Several authors indicate that tobacco use is determined by factors like perception, self-image, peer interaction, social norms, environment, and advertisement, which make it difficult to study and measure smoking influence, especially in Asian communities. Most consistent literature founding indicates the initiation of smoking is stress, coping, and personal construct related to peer interaction. For example, Eiser et al suggest that initiation is a driven behavior depending to which degree the adolescent comes in contact with peers displaying the same behavior. How this statement is correlated to the findings of the Korean population (Youth)?

Since the data get into account immigrant parent (family), what are the rate of prevalence between early emigration (before 2016 and before) compared to newly emigrated families (between 2016), some studies indicated that newly emigrated peer are more prone to develop smoking behavior. 

Some grammatical errors should be corrected. 

Author Response

Thank you for your review, which have helped us improve the quality of this manuscript. According to the reviewers’ comments, we have made revision on the manuscript. We used red colored fonts to indicate the revised parts for your recognition.

Reviewer 2 Report

- I suggest removing the "(KYRBS)(2016-2020)" from the title as this reports to the data set.

- line 36: which study? please cite.

- line 71: according to who? please add citations.

- line 116: smoking degree? please revise in the entire manuscript that smoking is a behavior and not a degree unless the authors explore this behavior in a categorical manner.

- Describe in detail how data was collected for applicability of future research.

- The measurement section (2.3., 2.4., and 2.5.) is very difficult to read. Please transform the information into a table.

- report cutoffs, values of reference, or the reported data in each data analysis test for clarity and transparency.

- provide a priori sample size calculations for transparency.

- limitations of the study should be disclosed.

- practical implications should be provided.

Dar Editor, thank you for inviting me to review this study. Consider my comments for your editorial decision.

Author Response

Thank you for your review, which have helped us improve the quality of this manuscript. According to the reviewers’ comments, we have made revision on the manuscript. We used blue colored fonts to indicate the revised parts for your recognition.

Round 2

Reviewer 1 Report

The authors have clarified several of the questions I raised in my previous review. Most of the problems have not been addressed by this revision. The tables and experimental design are well-designed. The manuscript is now warranted for publication with minor text editing and grammar corrections. 

This paper needs some text editing and grammar modification. for example running sentences (lines 63 to 66), extra wording, and wrong punctuation marks (line 77, comma instead of period).   

Author Response

(The authors gave the same response as above.)

Reviewer 2 Report

The authors did a good job reviewing their manuscript. I do not have further comments.

Dear Editor, I do not have further comments.

Author Response

Thank you for your review, which have helped us improve the quality of this manuscript.